# The Role of Traditional Acupuncture in Patients with Fecal Incontinence—Mini-Review

**DOI:** 10.3390/ijerph18042112

**Published:** 2021-02-22

**Authors:** Agne Sipaviciute, Tomas Aukstikalnis, Narimantas E. Samalavicius, Audrius Dulskas

**Affiliations:** Faculty of Medicine, Vilnius University, LT-03101 Vilnius, Lithuania; agne.sipaviciute.vu.mf@gmail.com (A.S.); Tomas.aukstikalnis@santa.lt (T.A.); narimantas.samalavicius@gmail.com (N.E.S.)

**Keywords:** acupuncture, electroacupuncture, moxibustion, faecal incontinence, diarrhea, irritable bowel syndrome, bowel dysfunction

## Abstract

Objective: Fecal incontinence affects up to 15% of the general population, with higher rates of incidence among women and the elderly. Acupuncture is an old practice of Traditional Chinese Medicine that might be used to treat fecal incontinence. The aim of this mini review was to assess the effect of acupuncture for fecal incontinence. Materials and Methods: Cochrane Library, Web of Science, Embase, PubMed, and CENTRAL electronic databases were searched until August 2020. The following keywords were used: acupuncture, electroacupuncture, moxibustion, fecal incontinence, diarrhea, irritable bowel syndrome, and bowel dysfunction. In addition, references were searched. Five studies (two randomized controlled trials), out of 52,249 predefined publications after an electronic database search, were included into the review. Results: Overall, 143 patients were included. All studies report significant improvements in continence, although they all apply different acupuncture regimens. Randomized controlled trials show significant differences in experimental groups treated with acupuncture in improving continence. Significant improvement in quality of life scores was reported. In addition, improvement in fecal continence remained significantly improved after 18 months of follow-up. Conclusion: Acupuncture is a promising treatment alternative for fecal incontinence. Based on small, low-quality studies, it might be a safe, inexpensive, and efficient method. However, more high-quality studies are needed in order to apply this treatment technique routinely.

## 1. Introduction

Fecal incontinence (FI) can be defined as involuntary loss of flatus and/or liquid or solid stool [1]. In the general population, it affects up to 15% [1]. Rates depends on patients’ age and gender, more frequently affecting females, up to 40% in elderly women and even higher rates in specific groups with co-morbid conditions [2]. The incidence range is wide, and it is believed that the prevalence rate is underestimated, as less than 25% of patients report this complaint to their physicians, leaving a high number of cases undetected and unreported. Fecal incontinence is a debilitating condition, as it might occur at socially unacceptable situations, leading to embarrassment, and it causes hygiene problems, a considerable impact on sexual life, and social stigma formation, altogether impairing the quality of life (QOL), which might be defined as the overall comfort, happiness, and well-being experienced by an individual.

Treatment options include conservative methods, covering personal hygiene, diet control, pharmacological therapy, physiotherapy techniques for pelvic floor muscles, percutaneous tibial nerve stimulation (PTNS), transanal irrigation (TAI), and anal plug usage; minimal invasive means, including sacral neuromodulation (SNM), antegrade irrigation, anal radiofrequency (SECCA), and intrasphincteric injections; and surgical approaches, including sphincter repair, graciloplasty, artificial sphincter, and colostomy. The means of treatment and alternative methods of traditional medicine such as acupuncture still need to be reviewed [3]. 

Acupuncture is an important method of Traditional Chinese Medicine that is used for various medical purposes for over 2500 years. According to Chinese medicine, all healthy human body structures are in balance of Yin and Yang powers. Therefore, when there is a misbalance of interacting vital substances, and most importantly in the Qi energy, malfunctions and diseases occur. For example, while treating fecal incontinence, acupoints representing the kidneys have to be stimulated, since it is responsible for both urination and defecation processes [4]. Functional activity of vital Qi power can be controlled and restored through the application of needles, seeds, moxa, or beads to specific point on skin. 

The mechanism of acupuncture is not well described and thoroughly investigated but it is believed that acupuncture works through central nervous system–gut neuromodulation. Takahachi discussed the physiological and anatomical pathway of acupuncture on bowel motility [5]. The nucleus of the tractus solitarius, which receives sensory information from both gastrointestinal system and skin mechanoreceptors, is next to the vagal nucleus, both of which are involved in the vago-vagal reflex [6]. Moreover, the neurons of the nucleus of the tractus solitarius connect to the rostral ventrolateral medulla (RVLM), which projects to the intermediolateral nucleus in the spinal cord through the symphatetic preganglionic neurons [7,8]. During acupuncture, the nucleus of the tractus solitarius is stimulated, and this leads to the activation of the autonomic nerve function through the RVLM and/or the dorsal vagal complex. A study by Zhu et al. [9] applied acupuncture treatment for functional diarrhea and used functional magnetic nuclear imaging to demonstrate changes in particular brain regions and gastrointestinal function. The authors discussed that acupuncture techniques might improve the pathological brain activity related to the bowel motility through the modulation of the homeostatic afferent processing network.

Stimulation of somatic afferents induces both the pain-relieving effect through the release of neuropeptides and the increase in peripheral circulation through the activation of depressor afferents and blood pressure reduction. Perineal region acupuncture through the afferent stimulation of sciatic nerve induces improvements in sphincter tone and activity, inhibits gastrointestinal motility, and thus, often reduces incontinence [10]. 

In Western countries and their medicine, this method is considered “new” since it is very rarely used as a treatment method and usually not even available as an option. Over time, acupuncture has evolved along with technology, and nowadays, might be used in the combination with ultrasound, laser, infrared radiation and other means in order to achieve better results, although the efficacy of this method is very little investigated and scientific evidence is scarce. However, recently, we showed that acupuncture is an effective treatment modality for patients with long-term low anterior resection syndrome [11].

The intention of this article was to review current studies published on fecal incontinence treated by alternative and less frequently applied method of treatment—acupuncture. We aimed to evaluate the overall effect of acupuncture and efficiency compared to other means of treatment.

## 2. Materials and Methods

This systematic review was performed in agreement to the Preferred Reporting Items for Systematic Reviews and Meta-Analyses (PRISMA, Oxford, UK) statement [12].

### 2.1. Literature Search and Inclusion Criteria

Electronic databases of PubMed, Ovid MEDLINE, Ovid Embase, Web of Science, the Cochrane Central Register of Controlled Trials (Cochrane Library), CENTRAL, and the Allied and Complementary Medicine Databases (AMED) were utilized independently for conducting electronic searches until August 2020. The search term consisted of two parts: intervention method and disease: (“electroacupuncture therapy” or “acupuncture” or “laser acupuncture” or “warm acupuncture” or “electroacupuncture” or “acupuncture therapy” or “moxibostustion” or “laser acupuncture therapy” or “warm acupuncture therapy”) and (“fecal incontinence” or “FI” or “encopresis” or “defecation incontinence” or “diarrhea” or “bowel dysfunction”).

Randomized controlled trials, pilot studies, and clinical commentaries that were available in the English language were eligible for inclusion. Patients diagnosed with fecal incontinence regardless of its etiology or participants’ age, gender, and ethnicity treated with various regimens of acupuncture were included into this review. Animal studies and papers written in languages other than English were excluded.

This article reviews all studies published in English between January 2003 and August, 2020. Moreover, the reference lists were searched. Data from the chosen studies were extracted and pre-tested to assess their eligibility.

The inclusion criteria for this systemic review were: (1) original studies; (2) studies that analyzed fecal incontinence treated with acupuncture.

### 2.2. Definition of Fecal Incontinence

Fecal incontinence (FI) is defined as a "recurrent uncontrolled passage of fecal material for at least one month, in an individual with a developmental age of at least 4 years” [13]. Most common conditions leading to impaired continence of feces and/or gases are as follows: pelvic floor dyssynergia, irritable bowel syndrome, infections, metabolic causes, dementia, psychosis, obstetric trauma, neuropathy, spinal cord injury, and multiple sclerosis. These causes can be grouped into functional, sphincter weakness and sensory loss [3]. 

There are several common scales used to assess the severity of fecal incontinence, including the Vaizey Incontinence Score [14], Fecal Incontinence Severity Index (FISI) [15], Cleveland Clinic Fecal Incontinence Score/Wexner (CCFIS) [16], Visual Analogue Scale (VAS) [17], and the Fecal Incontinence Quality of Life Scale (FIQL) [18]. 

Anorectal manometry is the objective approach used to evaluate the effect of fecal incontinence treatment. It measures both resting pressure and the pressure during squeeze [19]. 

Independent authors (A.D. and E.S.) evaluated the quality of the included studies using Robins-I [20] GRADE [21] tools.

## 3. Results

### 3.1. Search Results

The electronic search using the predefined keywords resulted in 52,249 publications. Duplicates and studies not related to FI or acupuncture treatment were excluded, leaving 20 studies to consider. Sixteen fully available publications remained after eliminating papers written in languages other than English and unavailable articles. Nine articles were excluded after reading the published abstracts, as they did not meet the inclusion criteria and were not suitable for the topic of this review. This resulted in seven articles eligible for the topic to be reviewed; however, after excluding two commentaries, five studies were included into the final review: three pilot studies and two randomized controlled trials (Figure 1 and Table 1).

### 3.2. Effect of Acupuncture Treatment

Overviewed studies reported a statistically significant improvements in fecal incontinence symptoms, assessing them with the use of incontinence scales and comparing manometry results before and after acupuncture treatment. Moreover, none of the studies reported adverse effects. The summary of relevant studies included into the review is presented in Table 1. Overall, 143 patients were included.

All 40 patients diagnosed with copracrasia were randomly assigned into two groups in a randomized controlled trial by Zhao et al. [22]. Patients in the experimental group were treated with acupuncture-moxibustion applications three times a week for the first eight weeks, followed by two sessions a week for another four weeks, resulting in 32 warm acupuncture sessions in total. The control group received symptomatic treatment, support therapy, prevention and treatment of complications. The results showed that in the experimental group, the Vaizey incontinence score improved statistically significantly (*p* < 0.05) after 12 weeks of treatment and in the follow-up period compared to the control group. Self-rating scores for satisfactions were higher in the acupuncture group in comparison to the medication group. The effective rate in the experimental group was 80%, which was statistically different (*p* < 0.05) compared to the 50% of control group after the received treatments. However, the difference in the effective rate in the follow-up period was not statistically significant between the groups, which was 90% and 80%, respectively [22]. Franco et al. [10] in the pilot study reports statistically significant improvements in VAS scores assessing fecal incontinence symptoms (*p* < 0.0001) and general FIQL value (*p* < 0.05), where domains of embarrassment and life style changed the most after 10 weekly traditional acupuncture sessions and 61% of patients reported improved quality of life in every assessed domain. Fecal incontinence did not improve for only one patient according to VAS assessment, indicating that VAS values decreased for the remaining 94% of patients described in the study. In addition, 38% of patients reported no symptoms of incontinence after the treatment. All 18 patients included into this study had a FI diagnosis for at least 6 months and had impaired continence function confirmed by anorectal manometry; furthermore, they have not had any history of surgery, trauma, metabolic diseases, neurologic disorders, or cognitive deficits that could have caused the incontinence. During the first acupuncture session, seeds were applied on the pre-selected acupoints for three days. The following weekly sessions lasted for approximately 40 minutes each using systemic needles applied to pre-selected points both dorsally and ventrally. A randomized controlled trial conducted by Allam et al. [23] investigated the effects of acupuncture on patients with moderate fecal incontinence caused by anorectal surgeries, such as hemorrhoidectomy, fistulectomy, or fissurectomy. The experimental group (*n* = 20) underwent acupuncture treatment and performed pelvic floor exercises, while the control (placebo) group (*n* = 20) underwent a sham laser acupuncture and performed pelvic floor exercises. Infrared laser treatment was applied a month after the surgery three sessions per week for 4 weeks, 12 sessions in total. There were no significant differences in participant age or distribution of sexes between the groups. Moreover, the values of anorectal manometry and FISI did not differ statistically significantly between groups before the treatment. After 4 weeks of treatment, there was a significant improvement in both resting anal pressure and squeeze anal pressure in the experimental group, with the percentage improvement of 18.55% and 11.87% (*p* < 0.001) respectively, 10.49% and 6.62% (*p* > 0.001) in the control group. FISI values statistically significantly decreased in both groups: 26.44 (*p* > 0.001) decrease in the experimental group and 13.90% (*p* > 0.001) in the control group. Values in anorectal manometry and FISI improved statistically significantly in the experimental group in comparison to the control group (*p* > 0.001) [23]. Scaglia et al. [24] published a pilot study describing 15 females diagnosed with FI for a period longer than a year. All patients were treated with 10 weekly manual acupuncture sessions, and for six of them, for whom the achieved effect was insufficient, the treatment was followed by seven more sessions once a month. The standard anorectal manometry showed an increase in the resting anal pressure from 25 mmHg to 36 mmHg (*p* = 0.05) following the acupuncture. This was well retained in the follow-up period of 18 months. The ability to sustain the squeeze pressure significantly increased from 41 mmHg to 60 mmHg (*p* < 0.05) after the applied sessions; however, these manometry values returned to the pre-treatment level in the follow-up period. Reported results showed no improvement in the maximal sphincter squeeze pressure, rectal volume, and sensory function. Fecal incontinence symptoms evaluated with the use of a Wexner score reflected a statistically significant improvement in both the overall mean and median scores, from 10 to 0 (*p* < 0.05) and from 17 to 0 (*p* < 0.05), respectively. After the follow-up period, both overall mean and median scores of incontinence had a value of 1, which reflects a statistically important improvement [24]. Yang et al. [15] published a pilot study describing electroacupuncture application in 30 patients with cerebral or spinal injury, who experienced at least one period per day of urinary plus fecal incontinence episodes. Electroacupuncture was applied in the following manner: five daily sessions with two free days in between followed by two more sessions for uncured patients. Treatment was considered effective if a decrease of over than 50% of incontinence was achieved, while failure was reported if incontinence decreased by less than 50%. The results of the study demonstrated that 76.7% of participants were cured, for 6.7% of patients the electroacupuncture treatment was effective, and for the rest of the patients, 16.6%, the treatment failed. Comparing patients with spinal injury (*n* = 9) and participants with cerebrovascular diseases (*n* = 20), no statistically significant difference was found, with the cured rate being 66.7% and 80% respectively [15]. 

### 3.3. Evaluation According to the Criteria

The methodological and conceptual quality of the reviewed studies were assessed according to the Robins I and GRADE (Table 2) [13,14]. The evaluation showed that the risk of bias was moderate (one study) to serious (all the rest studies). Moreover, the quality of the included studies was poor (Table 3).

## 4. Discussion

Today, the most popular neuromodulatory technique to treat fecal incontinence is sacral nerve stimulation (SNS), which is an invasive surgical method—an electrode is permanently implanted to stimulate the sacral nerve. Studies report high effectiveness in continence and overall quality of life improvement after SNS treatment [10]. However, this method, as a surgical procedure, may cause complications such as wound infections, pain, or displacement of electrodes [24]. Moreover, SNS treatment is an expensive method—the total cost amounts to approximately 21,500 Euro, whereas the total costs of percutaneous electrical stimulation or acupuncture treatment would be 400 Euro, as manual acupuncture with needles does not require expensive equipment [21]. 

Acupuncture is a minimally invasive, safe method of treatment with no reported adverse effects. Usually, thin needles are inserted into the skin and stimulated manually or electrically during the session. The results are promising, as all reviewed studies report high effectiveness in improvement of fecal continence. Acupuncture treatment was effective in heterogenic groups of patients as well as on homogenous groups of patients with anorectal surgery or a history of spinal or cerebral injury. Evidence on the long-term effectiveness of acupuncture on fecal incontinence are scarce; however, an Italian study reporting results after 18 months of follow-up showed that the achieved effect lasts for at least 3 to 6 months [24]. 

Comparing different means of acupuncture, all studies reported significant improvements in continence, although there were practical differences [23]. Acupuncture protocols involve the stimulation of multiple points; this can easily be done simultaneously in manual acupuncture using needles, while in laser acupuncture, there is a single laser and optic fiber. Acupuncture is relatively safe and simple; thus, it can be performed by a trained physiotherapist or nurse. The achieved effect can be further maintained and prolonged with additional sessions. Even though acupuncture is not a routinely applied practice for fecal incontinence, patients should be informed of this alternative option [21].

Present studies on acupuncture treatment provides us with conclusions of limited credibility because of short-term follow-ups, small number of patients, and high variety of different acupuncture regimens [10]. Patients’ diaries converted into semi-quantitative terms, even though they are validated and approved by scientific societies, are of limited reliability [21]. Acupuncture treatment results are also evaluated with the use of anorectal manometry; however, sensory threshold, sphincter tone, and sustained squeeze pressure do not greatly reflect a good clinical improvement, as the anal contraction results return to the pre-treatment levels even though the incontinence scores remain significantly improved [24]. 

Although complications of acupuncture are rare, mainly limited to case reports, they still exist [25]. Most often, authors report infections (local and systemic) or internal organ or tissue injury, among others. 

Our study is limited by the small number of studies without comparing other techniques. Moreover, all the studies are very heterogeneous, with different protocols and scales used for incontinence assessment and treatment, which creates a risk of bias and poor quality. 

## 5. Conclusions

Acupuncture is a promising treatment alternative for fecal incontinence. Based on small, low quality studies, it might be a safe, inexpensive, and efficient method. However, randomized controlled studies assessing the quality of life and long-term results, comparing different acupuncture modalities and treatments are required.

## Figures and Tables

**Figure 1 ijerph-18-02112-f001:**
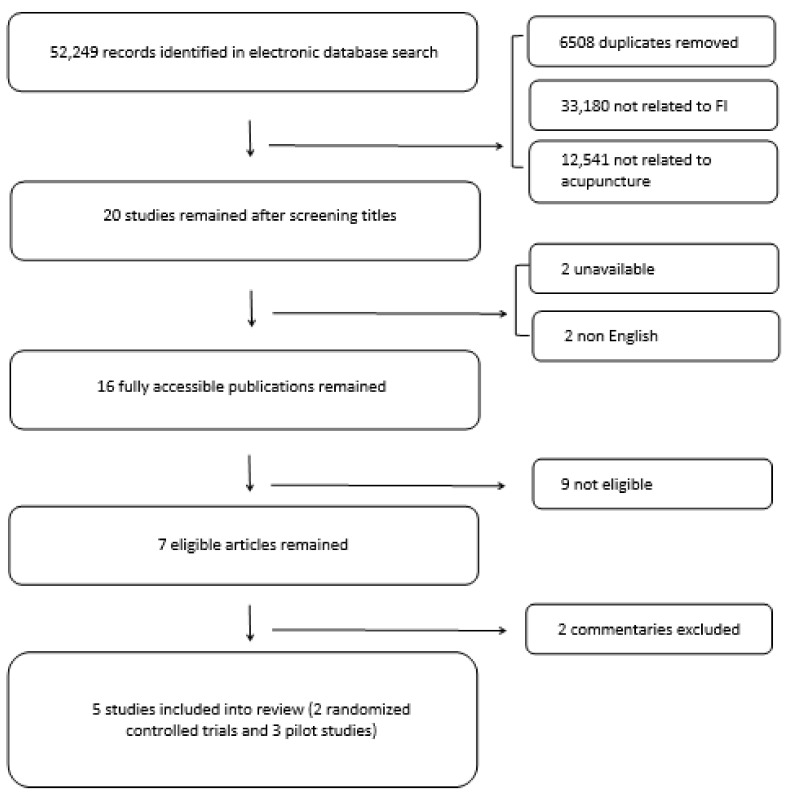
Flow chart of studies inclusion.

**Table 1 ijerph-18-02112-t001:** Summary of reviewed publications on fecal incontinence treated with acupuncture.

No.	Author and Year	Type	Patients Included	Follow-up	Inclusion Criteria	Exclusion Criteria	Treatment Technique	Acupoints	Diagnostic Tools for Incontinence Assessment	Results
1.	Allam N. et al. 2018	Randomized controlled trial	*N* = 40 *M* = 17 *F* = 23	-	Patients with moderate FI after anorectal surgery.	Patients with severe FI, damage to the rectovaginal fascia, spinal cord injury, neurological disorders, pregnancy, inflammatory bowel disease, idiopathic FI, history of FI before surgery, intake of photosensitive drugs, injury or active infection in the area of treatment, and unstable medical conditions.	Experimental group: laser acupuncture + pelvic floor exercises LA (infrared laser, 905nm, 15 W, 225 mJ, 60 s/point)—3 sessions per week for 4 weeks (1 month after the surgery). Control group: sham laser acupuncture + pelvic floor exercises	RN3, RN6, BL23, BL32, BL35, ST36, KI3	Anorectal manometry	18.55% increase in resting anal pressure and 11.87% increase in squeeze anal pressure after treatment in the experimental group (*p* > 0.001). 10.49% increase in resting anal pressure and 6.62% increase in squeeze anal pressure after treatment in the control group (*p* > 0.001). Significant increase in the experimental group compared to the control group after the treatment (*p* > 0.001).
									FISI	26.44% decrease in FISI in the experimental group after the treatment (*p* > 0.001). 13.90% decrease in FISI in the control group after the treatment (*p* > 0.001). Significant decrease in FISI in the experimental group compared to the control group after the treatment (*p* > 0.001).
2.	Franco J. et al. 2016	Pilot study	*N* = 18 *M* = 2 *F* = 16	-	Adult patients diagnosed with FI for more than 6 months and lack of continence function shown by anorectal manometry.	Patients younger than 21 years of age; prior muscle injury caused by surgical episiotomy or accident, neurological disorders, metabolic diseases, cognitive deficits, or other factors that could influence the incontinence, such as chronic diarrhea.	Traditional acupuncture, 10 weekly sessions. First session—seeds attached for 3 days on preselected points. Follow up sessions—needles applied to the dorsal area and held for 20 min, then to the ventral area for 15 min.	LI11, PC6, ST37, SP9, BL54, KI7, CV9 CV6, CV3, LV13, BL23, BL25, BL32, LU9, TH5, SP6, BL67, SP2, GB41	VAS	Statistically significant improvement (*p* < 0.0001).
									FIQL	Statistically significant improvement in each domain (*p* < 0.05).
										Adverse effects not reported.
3.	Scaglia M. et al. 2009	Pilot study	*N* = 15 *M* = 0 *F* = 15	18 months	Patients diagnosed with FI for longer than a year.		Manual acupuncture Once a week for 10 weeks, and once a month for 7 months for 6 patients.	3RM, 6RM, 4DM, 23BL, 32BL, 4LI, 36ST, 3K	Anorectal manometry	Resting anal pressure increased from 25 mmHg to 36 mmHG (*p* = 0.05); maximal sphincter squeeze pressure, rectal volume, and sensory function remained unchanged; the ability to sustain the squeeze pressure improved from 41 mmHg to 60 mmHg (*p* < 0.05) and returned to the pre-treatment values after the follow-up period.
									Cleveland clinic continence score/Wexner score	Significant improvement (*p* < 0.05) after treatment and at the final assessment in overall and mean scores.
4.	Yang T. et al. 2003	Pilot study	*N* = 30 *M* = 16 *F* = 14	-	Patients with at least one-time urinary incontinence plus FI episode per day caused by cerebral or spinal injury.	NA	Electroacupuncture (EA) 5 daily sessions with 2 free days between 2 courses + 2 more courses for uncured patients.	BL32, BL35	Effectiveness: Decrease of incontinence by over 50%. Failure: Decrease of incontinence by less than 50% or without improvement.	76.7% of FI patients were cured; for 6.7% of participants, the treatment was effective, while for 16.6% of patients, the treatment failed. No significant difference was found when comparing patients with a spinal injury and patients with cerebrovascular diseases, with a cured rate for FI being 66.7% and 80%, respectively.
5.	Zhao Y. et al. 2015	Randomized controlled trial	*N* = 40	NA	Patients with copracrasia.		Acupuncture-moxibustion group Acupuncture-moxibustion 3 times a week for 8 weeks, then 2 times a week for 4 weeks. Medication group. Symptomatic treatment, support therapy, prevention, and treatment of complications.	BL32, GV1, ST25, CV6	Vaizey incontinence score	Vaizey incontinence’ scores both statistically significantly decreased (both *p P* < 0.05) in acupuncture group after treatment and in the follow-up period compared to medication group.
									Effective rate	The effective rate of the acupuncture-moxibustion group was 80.0% (16/20), which was statistically different from 50.0% (10/20) in the medication group (*p* < 0.05). The effective rate in the follow-up period of the acupuncture-moxibustion group was 90.0% (18/20), which was not statistically different from 80.0% (16/20) in the medication group (*p* > 0. 05).
									Self-rating score for satisfaction	The self-rating scores for satisfaction in the acupuncture-moxibustion group were superior.

**Table 2 ijerph-18-02112-t002:** Risk of bias assessment of studies included in the review.

Domain	Allam N. et al.	Franco J. et al.	Scaglia M. et al.	Yang T. et al.	Zhao Y. et al.
Bias due to confounding	Serious	Serious	Moderate	Serious	Serious
Bias in selection of participants into the study	Low	Moderate	Moderate	Moderate	Low
Bias in classification of interventions	Low	Low	Low	Low	Low
Bias due to deviations from intended interventions	Low	Moderate	Moderate	Low	Moderate
Bias due to missing data	Low	Low	Low	Low	Low
Bias in measurement of outcomes	Low	Serious	Low	Serious	Serious
Bias in selection of the reported result	Low	Low	Low	Low	No information
Overall Risk	Serious	Serious	Moderate	Serious	Serious

**Table 3 ijerph-18-02112-t003:** Quality of studies included in the review.

Quality Assessment						Summary of Findings				Importance
						No of Patients		Effect	Quality	
No. of Studies	Study Design	Risk of Bias	Inconsistency	Indirectness	Imprecision	Acupuncture	No Intervention			
Incontinence										
5	RCT and Pilot	Serious	Not serious	Not serious	Not serious	103	40	Statistically significant improvement	Moderate	Important
Quality of life										
2	RCT and Pilot	Serious	Not serious	Not serious	Not serious	38	20	Statistically significant improvement	Moderate	Important
Anorectal manometry										
2	RCT and Pilot	Moderate	Not serious	Not serious	Not serious	38	20	Statistically significant improvement	Moderate	Important

## Data Availability

Data are available upon reasonable request.

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
