# Peer review of "The Role of Traditional Acupuncture in Patients with Fecal Incontinence—Mini-Review"

_ijerph, 2021, doi:10.3390/ijerph18042112_

Round 1

Reviewer 1 Report

The review titled: ‘The role of traditional acupuncture in patients with fecal incontinence – mini-review’ by Sipaviciute A et al. needs to be revised fot the English, especially some parts (see below). Further I would suggest the authors to pay attention to what they say in the discussion and in the conclusions. To me, the two parts are not coherent

Some concerns
MATERIALS AND METHODS
2.1
row 76, about the sentence: ‘Moreover, abstracts and reference lists were searched’ it is no clear the significance.
2.3
Row 103-105: the sentence: ‘Functional activity of vital Qi power can be controlled and restored through the application of needles, seeds, moxa, or beads to specific point on skin and thus the disease treated’. Needs to be better explained. Particularly the last part is not understandable.

Use always the term ‘nucleus of the tractus solitarius’ (no nucleus solitarius)
Row 121: write ‘induces’
Row 126: write ‘..and their medicine’.

RESULTS

3.1 It is impressive the enormous cut the authors made of the publications initially found. Likely their key words were not adequately chosen

Author Response

Dear Editor,

Thank you for your letter and constructive comments concerning our manuscript entitled “The role of traditional acupuncture in patients with fecal incontinence – mini-review”. The paper was revised substantially. Following changes have been made. They are as follows:

The review titled: ‘The role of traditional acupuncture in patients with fecal incontinence – mini-review’ by Sipaviciute A et al. needs to be revised fot the English, especially some parts (see below). Further I would suggest the authors to pay attention to what they say in the discussion and in the conclusions. To me, the two parts are not coherent

We have changed the conclusion part according to our results.

Some concerns
MATERIALS AND METHODS
2.1
row 76, about the sentence: ‘Moreover, abstracts and reference lists were searched’ it is no clear the significance.

Abstracts part was removed. We looked through the references of all the included studies aiming to find more related studies.

2.3
Row 103-105: the sentence: ‘Functional activity of vital Qi power can be controlled and restored through the application of needles, seeds, moxa, or beads to specific point on skin and thus the disease treated’. Needs to be better explained. Particularly the last part is not understandable.

The sentence was rewritten accordingly.

Use always the term ‘nucleus of the tractus solitarius’ (no nucleus solitarius)

Changed as suggested.

Row 121: write ‘induces’

Thank you for the correction!

Row 126: write ‘..and their medicine’.

Thank you for the correction!

RESULTS

3.1 It is impressive the enormous cut the authors made of the publications initially found. Likely their key words were not adequately chosen

Yes, we do agree that the decrease is significant. We have updated our search strategy – included in the Methods part.

Thank you very much indeed.

Reviewer 2 Report

Fecal incontinence affects up to 15% of the population and decreases the life quality of affected individuals. Therefore, it is essential to elucidate underlying disease mechanisms and to develop therapeutic strategies. In their mini-review, Sipaviciute et al. focused on literature describing the effect of acupuncture on fecal incontinence. While the topic will be of interest to the readership of the International Journal of Environmental Research and Public Health, several minor points need to be addressed prior to publication as detailed below:

The section “2.3. Description of acupuncture” should not be in the Materials & Methods section but rather part of the introduction because this is not a method that the authors applied in order to write their review.

Please increase the quality of Figure 1.

The manuscript contains several typing and grammatical errors which need to be corrected (e.g. “iincrease”, “thecontrol”). Moreover, the language used in a manuscript should be more neutral and professional (e.g. “[…] for at least 3 to 6 months what is a reasonable period of time.”).

Despite the fact that acupuncture rarely leads to side effects, the authors should briefly address the potential risk of this method (example: PMID: 23573135).

Author Response

Dear Reviewer,

Thank you for your letter and constructive comments concerning our manuscript entitled “The role of traditional acupuncture in patients with fecal incontinence – mini-review”. The paper was revised substantially. Following changes have been made. They are as follows:

The review titled: ‘The role of traditional acupuncture in patients with fecal incontinence – mini-review’ by Sipaviciute A et al. needs to be revised fot the English, especially some parts (see below). Further I would suggest the authors to pay attention to what they say in the discussion and in the conclusions. To me, the two parts are not coherent

The conclusions were changed according to Discussion and Results. Thank you for the suggestion.

Fecal incontinence affects up to 15% of the population and decreases the life quality of affected individuals. Therefore, it is essential to elucidate underlying disease mechanisms and to develop therapeutic strategies. In their mini-review, Sipaviciute et al. focused on literature describing the effect of acupuncture on fecal incontinence. While the topic will be of interest to the readership of the International Journal of Environmental Research and Public Health, several minor points need to be addressed prior to publication as detailed below:

The section “2.3. Description of acupuncture” should not be in the Materials & Methods section but rather part of the introduction because this is not a method that the authors applied in order to write their review.

Description on acupuncture moved from Methods to Introduction.

Please increase the quality of Figure 1.

The quality was increased.

The manuscript contains several typing and grammatical errors which need to be corrected (e.g. “iincrease”, “thecontrol”). Moreover, the language used in a manuscript should be more neutral and professional (e.g. “[…] for at least 3 to 6 months what is a reasonable period of time.”).

Changed as per suggestion.

Despite the fact that acupuncture rarely leads to side effects, the authors should briefly address the potential risk of this method (example: PMID: 23573135).

Short paragraph on complications inserted.

Thank you very much indeed.

Reviewer 3 Report

  • Please capitalize Fecal as the first word of your Objective section in abstract.
  • Similarly capitalize Acupuncture as the first word of your Conclusion section of abstract.
  • Various English grammar and capitalization corrects are needed.
  • The objective of the manuscript and study is clear. The authors are unfortunately hindered by the low volume of published articles on this topic. A very significant finding was noted by the authors in the high level of potential bias and low quality of the studies. Despite this very significant finding the authors conclude that acupuncture is safe, effective and cost-effective. This conclusion is overstated based on the methods and results.
  • Significant revision, mainly in the claims and conclusion are needed before acceptance. 

Author Response

Dear Editor,

Thank you for your letter and constructive comments concerning our manuscript entitled “The role of traditional acupuncture in patients with fecal incontinence – mini-review”. The paper was revised substantially. Following changes have been made. They are as follows:

The review titled: ‘The role of traditional acupuncture in patients with fecal incontinence – mini-review’ by Sipaviciute A et al. needs to be revised fot the English, especially some parts (see below). Further I would suggest the authors to pay attention to what they say in the discussion and in the conclusions. To me, the two parts are not coherent

  • Please capitalize Fecal as the first word of your Objective section in abstract.

Changed as suggested.

  • Similarly capitalize Acupuncture as the first word of your Conclusion section of abstract.

Changed as suggested.

  • Various English grammar and capitalization corrects are needed.

Manuscript was revised by the native English speaker

  • The objective of the manuscript and study is clear. The authors are unfortunately hindered by the low volume of published articles on this topic. A very significant finding was noted by the authors in the high level of potential bias and low quality of the studies. Despite this very significant finding the authors conclude that acupuncture is safe, effective and cost-effective. This conclusion is overstated based on the methods and results.

Conclusion were changed according the results. Thank you for great comment.

  • Significant revision, mainly in the claims and conclusion are needed before acceptance. 

The text was revised accordingly. Conclusion were changed according the results. Thank you for great comment.

Thank you very much indeed.

Reviewer 4 Report

Authors showed that acupuncture practice may be safe, efficient, cost-effective treatment approach for fecal incontinence improving both physical measures and quality of life scores. However, Further studies assessing the quality of life, long-term results, comparing different acupuncture modalities, and treatments are warranted. 

Author Response

Dear Editor,

Thank you for your letter and constructive comments concerning our manuscript entitled “The role of traditional acupuncture in patients with fecal incon-tinence – mini-review”. The paper was revised substantially. Following changes have been made. They are as follows:

The review titled: ‘The role of traditional acupuncture in patients with fecal incontinence – mini-review’ by Sipaviciute A et al. needs to be revised fot the English, especially some parts (see below). Further I would suggest the authors to pay attention to what they say in the discussion and in the conclusions. To me, the two parts are not coherent

Authors showed that acupuncture practice may be safe, efficient, cost-effective treatment approach for fecal incontinence improving both physical measures and quality of life scores. However, Further studies assessing the quality of life, long-term results, comparing different acupuncture modalities, and treatments are warranted. 

We thank you very much for your kind comment. We do agree that further studies are needed.

Thank you very much indeed.

Round 2

Reviewer 1 Report

I highly recommend a thorough review by a native English speaker. There are several sentences throughout the text that sound bad when reading. There are also repetitions that could be avoided by improving your English.

Reviewer 3 Report

Concerns were addressed. Could still improve English language some.